# Hybrid Neuro-Symbolic Reasoning based on Multimodal Fusion

## Abstract

Deep neural models and symbolic Artificial Intelligence (AI) systems have contrasting advantages and disadvantages. Neural models can be trained from raw, incomplete and noisy data to obtain abstraction of features at various levels, but their uninterpretability is well-known. On the other hand, the traditional rule-based symbolic reasoning encodes domain knowledge, but its failure is often attributed to the acquisition bottleneck. We propose to build a hybrid learning and reasoning system which is based on multimodal fusion approach that brings together advantageous features from both the paradigms. Specifically, we enhance convolutional neural networks (CNNs) with the structured information of '*if-then*' symbolic logic rules obtained via word embeddings corresponding to propositional symbols and terms. With many dozens of intuitive rules relating the type of a scene with its typical constituent objects, we are able to achieve significant improvement over the base CNN-based classification. Our approach is extendible to handle first-order logical syntax for rules and other deep learning models.

## 1 Introduction

Deep learning technology is being employed with increasing frequency in recent years LeCun et al. (2015)Schmidhuber (2014). Various deep learning models have achieved remarkable results in computer vision Krizhevsky et al. (2017), remote sensing Zhu et al. (2017), target classification in SAR images Chen et al. (2016), and speech recognition Graves et al. (2013)Hinton et al. (2012). In the domain of natural language processing (NLP), deep learning methods are used to learn word vector representations through neural language models Mikolov et al. (2013) and performing composition over the learned word-vectors for classification Collobert et al. (2011). Convolutional neural networks (CNNs), for example, utilize layers with convolving filters that are applied to local features. CNN is widely used for image tasks and is currently state-of-the-art for object recognition and detection. Originally invented for computer vision, CNN models have subsequently been shown to be effective for NLP and have achieved excellent results in semantic parsing Yih et al. (2015), search query retrieval Shen et al. (2014), sentence modelling Kalchbrenner et al. (2014), and other traditional NLP tasks Collobert et al. (2011).

However, the success of deep learning comes at a cost. The first and foremost is its reliance on large amounts of labeled data, which are often difficult to collect and entail a slow learning process. Second, deep models are brittle in the sense that a trained network that performs well on one task often performs very poorly on a new task, even if the new task is very similar to the one it was originally trained on. Third, they are strictly reactive, meaning that they do not use high-level processes such as planning, causal reasoning, or analogical reasoning. Fourth, human expertise cannot be used which can often reduce the burden of acquiring training data which is often expensive to collect. Purely data-driven learning can lead to uninterpretable and sometimes counter-intuitive results Nguyen et al. (2014)Szegedy et al. (2013).

The sub-symbolic neural approaches allow us to mimic human cognitive thought processes by extracting features at various levels of abstraction from direct observation and thereby facilitate learning. But humans also learn from general high-level knowledge expressed declaratively in logical syntax. A representation language allows recursive structures to be easily represented and manipulated, which is usually difficult in a neural learning environment. But a symbolic reasoning system is not good to adapt to new environments by learning and reasoning based on traditional theorem-

proving, which can be computationally expensive. Moreover, a purely symbolic system based on traditional AI requires enormous human effort as knowledge are manually programmed and not learned. Central to classical AI is the use of language-like propositional representations to encode knowledge. The symbolic elements of a representation in classical AI – the constants, functions, and predicates – are typically hand-crafted. Inductive Logic Programming Muggleton (1990) methods learn hypothesis rules given background knowledge, and a set of positive and negative examples. The systems we have discussed until now do not model uncertainty which is essential in practical applications. Various probabilistic logics Halpern (2005) and Markov Logic Networks Richardson & Domingos (2006) (MLNs) handle uncertainty using weight attached to every rule. Practical applications of these networks have been limited as inference is not scalable to a large number of rules.

It is, therefore, desirable to develop a hybrid approach, embedding declarative representation of knowledge, such as domain and commonsense knowledge, within a neural system. In this paper, hybrid approach is applied to indoor scene classification, which has been extensively studied in field of computer vision Chen et al. (2018). However, compared with outdoor scene classification, this is an arduous issue due to the large variety of density of objects within a typical scene. In addition, high-accuracy models already exist for outdoor scene classification while indoor scene classification is not. In order to accomplish our objective, the acquisition, representation, and utilization of visual commonsense knowledge represents a set of critical opportunities in advancing computer vision past the stage where it simply classifies or identifies which objects occur in imagery Davis et al. (2015).

The contributions of this paper is summarized as followed:

- A joint representation multimodal fusion framework is applied to exploit the early fusion of vectorized logical knowledge and images for the task of indoor scene classification. Experiments show that higher classification accuracy is obtained compared to traditional image classification methods.

- A '*if-then*' logical knowledge system is built based on reviews of each indoor scene class which are scraped from Google open source, through Word2Vec and BERT embedding. This helps to get a better contextual representation of words detected by object detection.

- A unique rules embedding approach is proposed, which allows to converge '*if-then*' logic of probability with image representation. The embedding approach has different representations during training and inference process.

The rest of the paper is organized as follows. The next section 2 surveys the related work. The hybrid framework is explained in section 3. Section 4 details implementation and evaluation of experiments. Finally, we conclude with some future directions in 5.

## 2 RELATED WORK

Hybrid neural-symbolic systems concern Chen et al. (2016)Garcez et al. (2009)Hammer & Hitzler (2007)Rosenbloom et al. (2017)Sun (1994)Wermter & Sun (2001) the use of problem-specific symbolic knowledge within the neurocomputing paradigm, specifically, symbolic domain and commonsense knowledge within the deep learning paradigm in our case. They are useful for enhancing various tasks, including logical inferencing, extracting relational knowledge Guillame-Bert et al. (2010)Gust et al. (2007), image classification, and action selection.

Combination of logic rules and neural networks has been considered to construct network architectures from given rules to perform reasoning and knowledge acquisition. Neural-symbolic systems, such as EBL-ANN Shavlik & Towell (1989), KBANN Szegedy et al. (2013) and C-ILP Garcez et al. (2009), LENSR Xie et al. (2019), like our proposal, deal with propositional formulae. KBANN, for example, maps problem-specific domain theories, represented in propositional logic, into neural networks and then refines this reformulated knowledge using back-propagation. Propositional symbols are directly represented as nodes whereas we vectorize each propositional symbol as its semantic representation and appended to the abstraction of low-level observations.

Other neural-symbolic systems are exploring on knowledge graph Chen et al. (2020)Kampffmeyer et al. (2019)Li et al. (2019)Zablocki et al. (2019), which is a natural symbol. It is not only a se-

mantic network to describe entity relationships, but also a formal description framework for general semantic knowledge.

A large amount of neural-symbolic approaches focus on first-order inference Zhang et al. (2020)Marra et al. (2020)Yang & Song (2020)Cai et al. (2021). But some do not allow one to learn vector representations of symbols from training facts of a knowledge base, such as SHRUTI Shastri (1999), Neural Prolog Ding et al. (1996), CLIP++ Franca et al. (2014), and Lifted Relational Neural Networks Sourek et al. (2015). Neural Reasoner Peng et al. (2015) translates query representations in vector space without rule representations and can, thus, not incorporate domain specific knowledge. The Neural Theorem Prover Rocktäschel & Riedel (2016) and Unification Neural Networks Hölldobler (1990)Komendantskaya (2011) build upon differentiable backward chaining, but the former operates on vector representations of symbols whereas the latter on scalar values. Grounded Abductive Learning (GABL) Cai et al. (2021) is proposed to enhance machine learning models with abductive reasoning in a ground domain knowledge base, which offers inexact supervision through a set of logic propositions.

Different frameworks of neural-symbolic system rely on various logics Raedt et al. (2019). For example, Logic Tensor Networks Donadello et al. (2017) is based on first-order logic, Lifted Rule Injection Demeester et al. (2016) exploits implication rules, Semantic Loss Function Xu et al. (2018) focuses on propositional logic. Other deep learning systems such as TensorLog Cohen (2016) uses datalog, while DeepProbLog Manhaeve et al. (2019) uses clausal logic. Among these frameworks, Semantic Loss Function and DeepProbLog not only embed logic, but also add probabilistic to neural networks, which are two principal factors of reasoning.

Like our method, DeepProbLog Manhaeve et al. (2019) purposes a framework where expressive probabilistic-logical modeling and reasoning are combined, which could be trained end-to-end. Yang et al. Yang & Song (2020) proposed Neural Logic Inductive Learning (NLIL), an efficient differentiable ILP framework that learns first-order logic rules to explain problem in the scope of inductive logic programming (ILP). Marra et al. Marra et al. (2020) presented Relational Neural Machines (RNM), a novel framework to converge deep architectures and probabilistic logic reasoning. It is able to recover both classical learning in case of pure sub-symbolic learning, and Markov Logic Networks in case of pure symbolic reasoning. Mao et al. Mao et al. (2019) proposed the neuro-symbolic concept learner (NS-CL), which is able to represent object-based scene and translate sentences into executable, symbolic programs.

## 3 HYBRID NEURO-SYMBOLIC METHOD

### 3.1 OVERVIEW

Logical knowledge representation is symbolic in nature, i.e. the data structures under consideration basically consist of words over some language. A logic program Lloyd (1984) is a set of (universally quantified) disjunctions, called clauses or rules, which in turn consist of atoms and negated atoms only. Definite rules of 'if-then' type have a conjunction of atoms as antecedent and one atom as consequent. Successful connectionist architectures, however, can be understood as networks (essentially, directed graphs) of simple computational units, in which activation is propagated and combined in certain ways adhering to connectionist principles. In many cases like, for example, in multi-layer perceptrons, the activation is encoded as a real number; input and output of such networks consist of tuples (vectors) of real numbers.

In order to integrate logic and connectionism, we, thus, have bridged the gap between the discrete, symbolic setting of logic, and the continuous, real-valued setting of artificial neural networks. In this paper, we propose an approach to multimodal learning which involves relating information from multiple sources. Specifically, we focus on learning representations for images which are coupled with vectorized features of propositional rules, along the line of multimodal video/audio deep learning in Ngiam et al. (2011) or a multimodal deep Boltzmann machine (DBM) Srivastava & Salakhutdinov (2012). The bimodal DBM in our case models the joint distribution over image and symbolic knowledge inputs. The joint distribution over the multi-modal input variables is written as

$$p(v^m, v^a; \theta) = \frac{1}{Z_M(\theta)} \times$$

$$\sum_h \exp(Replicated\ Softmax\ Symbolic\ Pathway$$

$$+ Gaussian\ Image\ Pathway$$
$$+ Joint\ Fully\ Connected\ Layer)$$

where $h$ is all hidden variables with superscripts in parentheses, $Z$ is the normalization constant depending on the number of propositional symbols in all the rules. The image-specific DBM uses Gaussian distribution to model the distribution over real-valued image features. Similarly, rule-specific specific DBM uses Replicated Softmax to model the distribution over word count vectors.

Word-embeddings of symbolic and subjective rules adopt divergent representations, whilst Word2Vec and BERT are introduced in this paper to represent the contextual knowledge. The usage of BERT representation will provide stronger semantic contextual meaning and relevance with each label, so that the hybrid network is able to understand symbolic rules better Kalchbrenner et al. (2014). The Word2Vec representation captures many linguistic regularities and preserves semantic similarity meaning. We set the value of the absent modality to zero when computing the shared representation, which is consistent with the feature learning phase. Word-embeddings of symbolic rules and objects images are extracted and input into the Word2Vec and BERT model.

The proposed hybrid framework with CNN has been applied to an indoor scene classification scenario with 5 classes, namely, library, museum, concert, church, mall. Here is how we prepare the labelled instances for training and text from an image corpus and a set of logical rules, as shown in 1. In the scenario, for example, for an input image corpus, the pixel representation of each image is fed into the hybrid model but coupled with zeros if no object is identified in the image, or the vectorized representation of the identified objects within the image; so if the image scene is of type "*library*" and if the only identified object is "*shelf*" then an abstraction of the image pixel array at a certain level is coupled with the word embedding of "*shelf*". The combined representation then becomes a training sample with the label "*library*" and propagated into a fully connected classification network.

On the other hand, if we have a domain propositional logic rule "if *shelf* & *table* then *library* (0.8)", then vectorization of "*shelf* & *table*" is coupled with zeros for the image modality. The combined representation also then becomes a training sample, but in this case the component corresponding to "*library*" of the 1-to-C coding of the label will have 0.8 instead of 1. This hybrid approach is an early fusion of modalities. Moreover, in order to integrate Word2Vec and BERT representations, logic rules are alternated into these two representations to get specific vectorization as explained above, then merged horizontally into a vector.

We have made use of publicly available images and textual blogs to generate hundreds of domain rules. Like the example above, automatically from applying Bayes' probability and then merged with additional hand coded domain rules. Each image is also optionally tagged with a number of objects. We can leverage on one or more of many existing object detection frameworks, such as Spatial Pyramid Pooling (SPP) He et al. (2014), OverFeat Sermanet et al. (2013), Multibox SSD Liu et al. (2016), Fast R-CNN Girshick (2015), YOLO Redmon et al. (2016), and Faster R-CNN Ren et al. (2015), and TensorFlow based Google object detection or Caffe tool. The hybrid framework strongly improves over the basic CNN as illustrated by examples in Figure 1, but much more details are in the implementation and evaluation section. To the best of our knowledge, this is a state-of-the-art approach to tightly integrate traditional '*if-then*' logic rules with CNN.

## 3.2 RULES GENERATION

Two different types of logical rules are generated. One is initiated by object detection algorithm and Bayes' theorem, another is manually preset by us. Our main approach is based on automatically generated rules from Bayes' theorem. However, we have compared the effect of two types of rules within fusion network in our experiments, 4 shows the detail.

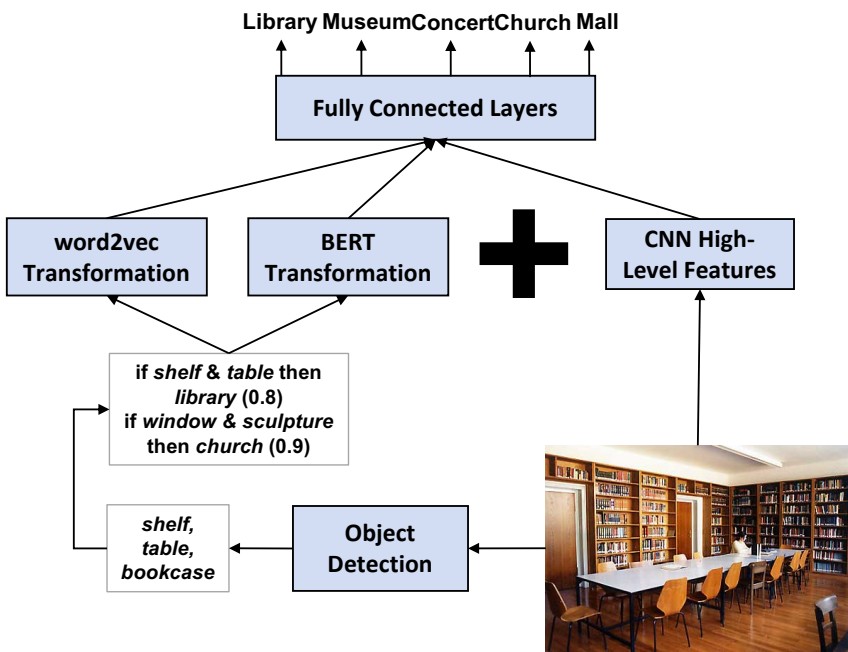

Figure 1: Hybrid architecture example

### 3.2.1 AUTOMATICALLY GENERATED RULES

For each image, through Fast R-CNN object detection algorithm Girshick (2015), we tag them with a number of objects anywhere of 10. For example, in a "*library*" image, a group of objects, "*chair, building, chair, chair, chair, shelf, shelf, table, couch, shelf*" is given. These 10 tags may contain repeated objects, so we implement the counting of repeatedly additions to figure out how many objects there are in each scenario. These objects which emerge less than 10 times in a scenario will be subtracted, since each scenario has 100+ images and the probability is too low to be counted which may influence the accuracy of the representation of image rules. Finally we get 2 to 8 objects each image. For example, an image of a "*library*" is likely to contain "*shelf, chair, table, bookcase*".

Combination theory helps to form rules subsets from single objects to multiple objects. $k$ objects are selected from a set of $n$ objects to produce subsets without ordering. The number of such subsets is denoted by $C_n^k = n!/(n-k)!$ where $k$ here is set as 2 and 3, which means that the combinations of 2 objects and 3 objects generate and will form rules of 2 features and 3 features. Same as the previous subtraction, these combinations which appear less than 10 times are removed to ensure avoid redundant feature rules generation. Thus, the probability of single objects or multiple object combinations can be calculated for each scenario, that is, the probability of "*shelf*", "*shelf & table*", "*shelf & table & bookcase*" emerge in a library image. Based on the Bayes' theorem, the probability of a certain scenario when an object or a combination of objects emerge can be calculated. For example, we use the theorem to calculate that if the combination of "*shelf & bookcase*" is in a scenario, the probability of "*library*" is 95%. Table 1 shows some image rules calculated using Bayes' theorem.

### 3.2.2 HUMAN ENCODED RULES

Within these artificially generated '*if-then*' logics, object keywords in the '*if-then*' logic are picked randomly by hand and are contextually relevant. The probability of each logic are set as follows: if an object appears alone, we haphazardly give the probability of 0.75 or 0.8; if it appears by a combination of two or three objects, then we add 0.05 or 0.1 at random to the probability of one or two objects correspondingly. Table 2 shows some logic rules initialized manually.

Table 1: Automatically Generated Image Rules Examples

| Prob | Class | Feature 1 | Feature 2 | Feature 3 |
|------|-------|-----------|-----------|-----------|
| 80 | library | shelf | | |
| 84 | library | chair | table | |
| 100 | library | shelf | desk | bookcase |
| 76 | museum | picture | | |
| 100 | museum | furniture | sculpture | |
| 100 | concert | curtain | | |
| 100 | concert | window | curtain | chair |
| 100 | church | bench | | |
| 74 | church | window | sculpture | houseplant |
| 100 | mall | vehicle | | |
| 82 | mall | person | building | plant |

### 3.3 MULTIMODAL NETWORK FRAMEWORK

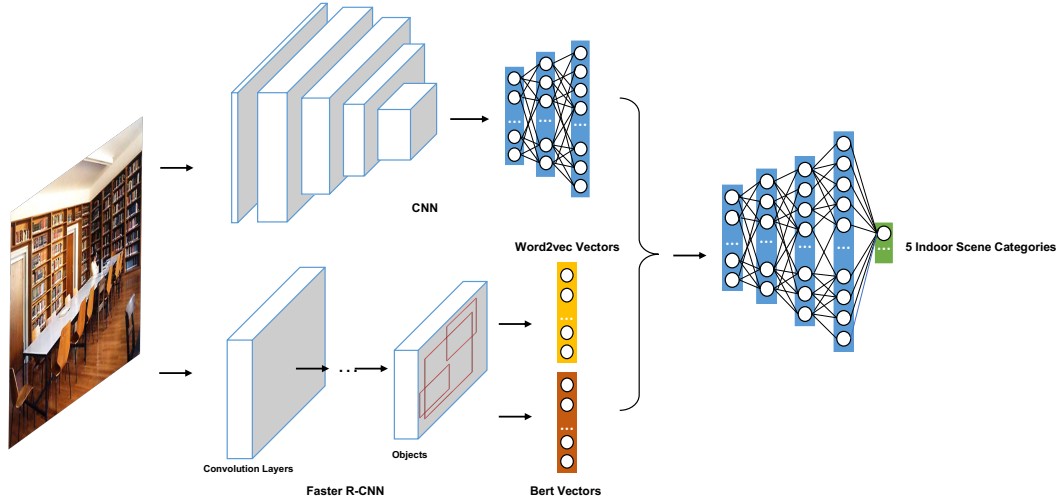

Figure 2: Multimodal Network Framework

We have constructed a multimodal neural network as shown in figure 2. Firstly, a 9-layer CNN model is utilized to extract representation of image features. Note that our main objective is not to come up with a best CNN architecture for scene classification but rather to show the relative improvement in the performance when the base architecture is hybridized with a symbolic approach.

Next, Faster R-CNN model is leveraged to implement object detections. 10 objects have been extracted from each image. We then generate several permuted 'if-then' rules based on these extracted objects through Bayes statistical method. These logical rules are represented by Word2Vec and BERT embeddings respectively. The pre-trained CBOW-based Word2Vec model is exploited to generate Word2Vec vectors for these rules. What's more, for engendering BERT vectors, the structure of BERT classifier is as follows: the small pre-trained BERT model from Tensorflow is loaded, followed by a fully connected layer with 5 units (number of classes) with a softmax activation function. After training, we extract intermediate vector as BERT representations of these knowledge.

After acquiring image features from CNN network, Word2Vec and BERT representations of logical knowledge rules, we perform multimodal fusion method on them. Specifically, we superimpose the dimension size of these image and rule vectors as the input dimension size of our subsequent fusion network. For each dimension of data, besides its own vector, we set zero values for these positions occupied by other two types of data. All the number of training image and probability-based training rule entries are concatenated for multimodal fusion training. However, when predicting,

we concatenate the image and its corresponding rules in the image into a vector as input, which is different from training. Obtaining joint representation of these multimodal features, a multi-layer perceptron is used to do fusion classification.

### 3.4 MODALITIES EMBEDDING

#### 3.4.1 TRAINING

For training, multiple label inputs for each image are formed as follows, which is divided into two parts. One is the image representation and the other is the rules embedding. The overall input dimension is 300+512+256. In terms of image features, the 300-d plus 512-d zero vectors are concatenated with 256-d last layer of CNN. For rules, the composite 300-d Word2Vec word-embedding and 512-d BERT word-embedding are formed using respective conditions in the antecedent of rules as is done for image tagging. Then, the vector is amplified with a 256-d vector of zeroes, which is used to align with the overall dimension to obtain a training instance for the rule. The label corresponding to this instance is the consequent of the rule. We replicate the rules instance proportional to associated confidence probability as needed to balance two sets of training instances generated from the raw images and rules. For example, if possibility of the rule is 100%, this instance is then replicated 10 times.

#### 3.4.2 INFERENCE

By including the individual word-embedding of the tagged objects, if any, of the image, the composite 300-d Word2Vec word-embedding and 512-d BERT word-embedding are then augmented with high-level features of 256-d for each labeled image are extracted from CNN. If there are no tagged objects with the image then the 300+512+256 dimensional input vector for the image is formed by padding with zeroes.

## 4 IMPLEMENTATION AND EVALUATION

### 4.1 DATASET

We have totally collected 733 images for all indoor scene classification scenario, almost equally distributed among the 5 classes "*library*", "*museum*","*concert*", "*church*" and "*mall*". Among 733 images, there are 440 used for training, 146 for validation and 146 for testing. The dataset is originally provided by MIT, which contains 67 Indoor categories, and a total of 15620 images. The number of images varies across categories, but there are at least 100 images per category. These images are of various dimensions and pixel resolution, but we have wrapped each to the fixed dimension 96x96.

### 4.2 NETWORK SETUP

In our proof-of-concept demonstration for hybrid learning, the CNN model as in 1 is implemented in TensorFlow. We have made use of Google's pre-trained Word2Vec model via the Python package genism. The dimension of the embedding vector is 300. The BERT model is pre-trained Tensorflow small English uncased BERT L-4_H-512_A-8 model. The text classifier is built on that which has a single neural layer using Softmax activation. The dimension of the embedding vector is 512.

The following parameters are used for training the multimodal hybrid network with rules transformation: Number of epochs = 50; Batch size = 64; Input size = 300 + 512 + 256; Learning rate = 0.001. Once the training of the hybrid model is completed, testing for each image from test dataset is done with an input obtained by concatenating extracted 256-d features from the image using the CNN with the composite 300-d Word2Vec word-embedding and 512-d BERT word-embedding of the identified objects in the image.

### 4.3 RESULT

we have provided additional knowledge of objects in input images into the hybrid network incorporating those domain rules. The hybrid network has both Word2Vec and BERT transformation

word-embedding. After the entire framework is trained and validated on the selected dataset, quantitative results are obtained from the evaluation on the testing set. Additionally, VGGnet16 and MobileNet V3 is referenced as the baseline model. The experiment results are show in 2. It shows the overall classification performance of hybrid network with two word-embeddings for the testing set and of the other two baseline networks. Across the board, the whole figure indicates the overall improvement of all metrics from VGGnet16 and MobileNet V3 to hybrid network, which adequately demonstrates the effectiveness of our framework. Note that for each of tagged object, which is imported as inputs to the network, there is at least one rule with one of the conditions in the antecedent matching the object. What's more, qualitative results for some images compared to baseline in the testing set are visualized in 3.

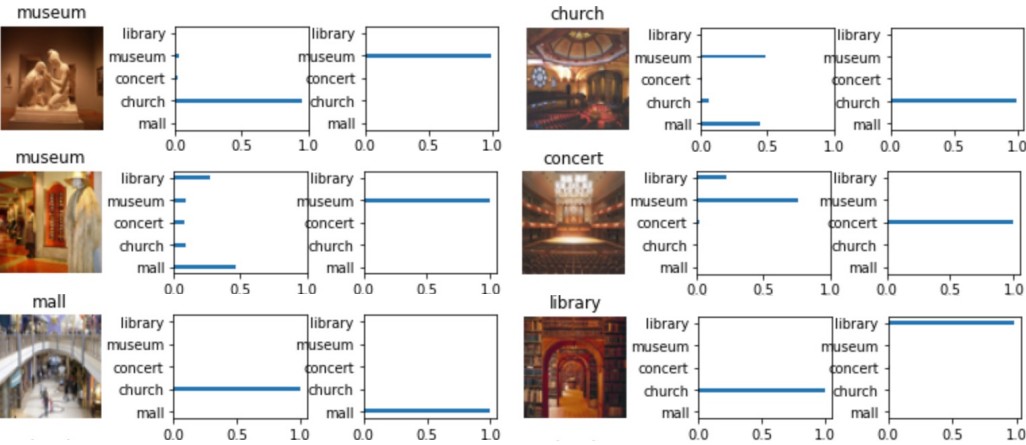

Figure 3: Overall improvement in prediction through base CNN to rules embedding

Table 2: Main evaluation results and baseline results

| Model | Accuracy(%) | Recall(%) | Precision(%) | F1(%) |
|---|---|---|---|---|
| VGG16 | 43.15 | 42.75 | 40.41 | 37.17 |
| MobileNet V3 | 37.67 | 40.77 | 36.30 | 34.27 |
| Ours | 73.29 | 74.31 | 73.29 | 73.02 |

## 4.4 PERFORMANCE EVALUATION

**First**, we tested the hybrid model to show that its functions like the basic 3-layer CNN when for each input image's symbolic part is reduced to zeros. This means for an input scene with no tagged identified objects, the base CNN will function the same way as the hybrid with no additional symbolic knowledge and it did in our case.

**Second**, we compared fusion of CNN image features plus automatically generated rules and fusion of CNN image features plus manual rules. The comparison results are shown in 3. As we can see, rules taken out from objects detection are better than the human encoded ones. This reveals that the former has more prior knowledge based on objects in each image, while the latter has more information not be restrained in the image.

Table 3: CNN with different kinds of rules

| Framework | Accuracy(%) | Recall(%) | Precision(%) | F1(%) |
|---|---|---|---|---|
| CNN + Human Encoded | 64.38 | 65.03 | 63.70 | 64.61 |
| CNN + Automatically Generated | 73.29 | 74.31 | 73.29 | 73.02 |

**Third**, we tested that only logic rules are used to predict the classification of indoor scenes. As shown in 4, our fusion framework has overall better performance than rules alone. Though, human

encoded rules fused with CNN have more performance gap to rules training only. This also reveals that these automatically generated rules from images can be regarded as a type of extracted feature.

Table 4: Only logic rules

| Rules | Accuracy(%) | Recall(%) | Precision(%) | F1(%) |
|---|---|---|---|---|
| Human Encoded | 34.93 | 34.93 | 34.93 | 26.96 |
| Automatically Generated | 73.29 | 73.29 | 73.29 | 68.37 |

**Fourth**, we have implemented three ablation experiments which are 3-layer CNN, 3-layer CNN plus Word2Vec representation and 3-layer CNN plus BERT representation. 5 shows the comparison results with our main hybrid fusion approach. It indicates that rules representation adding to image features helps to improve classification performance. BERT representation has more influence to classification results since the BERT classifier has been trained and represents word vectors better. Futhermore, the performance of our main experiment is beyond the fusion of representations of any single type of rules vector.

Table 5: Ablation study with different word representations

| Framework | Accuracy(%) | Recall(%) | Precision(%) | F1(%) |
|---|---|---|---|---|
| CNN | 56.16 | 57.34 | 56.16 | 53.04 |
| CNN + Word2Vec | 66.44 | 67.36 | 66.44 | 64.02 |
| CNN + BERT | 69.18 | 69.18 | 69.18 | 68.76 |
| CNN + Word2Vec + BERT (Ours) | 73.29 | 74.31 | 73.29 | 73.02 |

## 5 FUTURE WORK

We have developed a hybrid modeling framework which combines CNN with propositional rules to allow integrating subjective domain knowledge, relating objects to scenes, into the neural models. Our experiment with a scene classification scenario shows substantial improvement over classification by the underlying CNN model without rules. We are currently in the process of extending the hybrid modeling framework with rules expressed in Horn clauses which is a subset of full first-order formulae Lloyd (1984). We are also investigating how to incorporate uncertainty associated with the output of a typical object detection algorithm.

One application we are actively working on is situation and threat assessment Martin Liggins II (2008) within an area under surveillance, where intelligence comes from a variety of multimodal sources, including human and signal intelligence. Unlike a scene, which is visible, a specific situation or threat is a non-visible abstraction of various visible objects present in the environment. We believe subjective knowledge from a domain can help improving such assessment tasks.

Finally, Baker et al. Baker et al. (2018) observed that deep CNNs have access to some local shape information in the form of local edge relations. Tasks such as automated scene classification and situation assessment require learning of the presence or absence of objects and their spatial relationships. Hence, the scope and usefulness of incorporating domain expert knowledge and known sensor behavior into deep CNNs is very high.

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
