# OpenReview forum: "Hybrid Neuro-Symbolic Reasoning based on Multimodal Fusion"
_ICLR.cc/2023/Conference — Submitted to ICLR 2023_

### Official Review · Reviewer_vv79 · 2022-10-24

**Confidence:** 4
**Correctness:** 2
**Technical Novelty And Significance:** 2
**Empirical Novelty And Significance:** 2
**Recommendation:** 3

**Clarity, Quality, Novelty And Reproducibility:**

The clarity and quality of this paper are poor, the novelty of the proposed approach is limited, and the reproducibility is weak because codes and many details about the experiments are unavailable.

**Strength And Weaknesses:**

### Strengths:
- This paper targets at solving an important problem, which is highly related to this conference.

### Weakness:
- The presentation of this paper can be significantly improved, there are many grammatical errors that need to be revised. The only one equation is not numbered and is unexplained, and many descriptions are confusing. Apparently, this work needs proofreading before submission and is not ready for publication now.
- The authors claim that the proposed approach aims at bridging neural and symbolic models, however, the symbolic representation is converted into embeddings again. Since relational features are extracted by a pretrained fast RCNN, why not directly attach the RCNN to the final predictive MLP? What is the purpose of extracting combinations of objects as rules?
- The hyper-parameters on rule confidence seem arbitrary.
- The experiments are weak, no state-of-the-art scene classification models were compared.
- There are some works trying to learn visual concepts using combinations of sub-concepts in a symbolic form, for example:
  - Zhongyi Han, Le-Wen Cai, Wang-Zhou Dai, Yu-Xuan Huang, Benzheng Wei, Wei Wang, Yilong Yin, “Abductive Subconcept Learning.” Science China Information Sciences, 2022.

**Summary Of The Paper:**

This paper proposes to combine symbolic representation and distributed representation together to improve image classification. The proposed model first uses a pre-trained fast RCNN to perform object detection on every example, and then automatically construct a rule set, in which every rule is a combination of the discovered objects in one example. Then, these rules are used for extract relational features from training examples, which will be fed into a seq2seq deep learning model. Combined with the CNN features, the final feature vector will be further processed by an MLP to make the final prediction.

**Summary Of The Review:**

This paper in its current condition is not ready for publication.

---

### Official Review · Reviewer_XLX8 · 2022-10-24

**Confidence:** 4
**Correctness:** 1
**Technical Novelty And Significance:** 2
**Empirical Novelty And Significance:** 1
**Recommendation:** 1

**Clarity, Quality, Novelty And Reproducibility:**

**Clarity**: The text is somewhat readable, but it is not well structured and contains a number of grammatical errors.  The text, in my opinion, is not yet of publication quality.  The mathematical formalization of the proposed approach is severely lacking (i.e., most aspects of the model are not formalized at all).

I strongly recommend the authors to clearly separate between method description (in its own Section), where a proper formalization of all components of the method should be given (clearly describing each element - the object, the image embedder, the rule embedder, the classifier - formally and specifying the intended inputs and outputs), from data separation and experimental setup (in a different, later Section).

I also suggest the authors to fix the citation style.  It is sufficient to replace \cite{} or \citet{} in the text with \citep{}.

**Quality** The idea of using word embeddings of rules as input to a neural predictor might have some merit - in that it propagates information about what objects have been detected in a scene and what predictions are more likely.  In principle, this is fine.

The main issue with the paper is the evaluation, which is carried out on a single data set against only basic baselines.  Considering that this is an image classification task, I would have expected, at the bare minimum, a comparison against state-of-the-art ViTs, not against VGG16 (published in 2015) and MobileNet V3 (definitely newer, but specifically designed to trade off performance for size).  Notice that these baselines are rule-agnostic, meaning that any nesy approach (which makes use of the extra information provided by the rules) is unfairly advantaged.  A better option would have been to compare against SOTA neuro-symbolic strategies.  A comparison against at least Logic Tensor Networks is definitely doable, considering that LTNs have been evaluated in the context of object-relation recognition.  The ablation experiments are a good addition, but they are not enough to make the empirical results compelling.

All in all, I think that the empirical evaluation needs to be improved substantially.

**Novelty** There may exist early works on NeSy integration using feature fusion to take rules or concepts into account, but I cannot find the right references.  However, I think that the specific combination of object detection, text embeddings, and feature fusion proposed here is novel.

The related work section is quite exhaustive, in that it cites a number of relevant papers.  However, their relationship with the proposed approach is often not mentioned.  I urge the authors to streamline and clarify the related work, focusing on what approaches can and cannot considered direct competitors, and why.

**Reproducibility**  Implementation details of the empirical evaluation (e.g., choice of hyperparameters) are sparse and insufficient for reproducing the experiments.  I could not find the source code in the supplementary material, and I don't think the authors promised to release the code upon acceptance.

**Strength And Weaknesses:**

PROS
- Proposed method outperforms basic image models on the selected data set.
- Ablation experiments.

CONS
- Text is readable, but contains a number of grammatical mistakes.
- Method description not clearly distinguished from experimental details.
- Model is not formalized properly.
- Very limited empirical comparison on a single task against non-SOTA, non-NeSy baselines.
- Related work is quite comprehensive but poorly structured.
- Overall, the work generally feels incomplete and rushed.

**Summary Of The Paper:**

The authors introduce a neuro-symbolic image classifier based on feature fusion.  The idea is to glue together (1) the image representation provided by embedding layers with (2) BERT and Word2Vec embeddings of (a textual representation of the body of Horn) rules that match objects detected in the image, and then (3) feed the resulting vector to a neural classifier.  The two embedding approaches are chosen so as to capture different types of information, namely structural and semantic/conceptual, intuitively complementing each other.  The proposed approach is compared against VGG16 and MobileNet V3 on a single indoor scene classification data set.

**Summary Of The Review:**

The intuition behind the method might have some potential, but the text is not up-to-par and the experiments are severely lacking.

---

### Official Review · Reviewer_jnEW · 2022-10-25

**Confidence:** 3
**Correctness:** 3
**Technical Novelty And Significance:** 2
**Empirical Novelty And Significance:** 2
**Recommendation:** 5

**Clarity, Quality, Novelty And Reproducibility:**

Clarity: mostly OK.

Novelty: weak

{Quaity: avg

the paper seems to have been written in 17 (eg,Intro

Novelty maybe the bayesuan attributes



























































































































































































































































Clarity, mostly ok

Quality, needs more arguments and more results

Novelty mostky the use of probabilistic rules

 And Reproducibility Bad










**Strength And Weaknesses:**

Strengths
 it merges important ideas
the results are good.
Weaknesses¨:
Why? Often decisions are not clearly explained, eg Bert +/vs w2vec
The probabilistic rules deserve an example.
There is little info on how the system is configured


**Summary Of The Paper:**

This paper proposes a complex design to classify   Images: it   involves w2v, bert, user defined and bayesian features, that are sent to DNN,,
Results look good., but on the author'ś data.


**Summary Of The Review:**

The paper needs to give us a more formal presentation and evaluation.

More in detail;the key idea of the paper consists in feeding high--level attributes to a DNN. This leads the authors to a 3 layer arch:

1. The  first step is to run  a cnn and Fast-Run CNN to obtain high-level features from the images. This is followed by a filtering step. It is not clear if the parameters were chosen ad-hoc or if  there was some kind of search, It is also not clear whether the new features are fed to the DNN, or just used by step 2.

This is also not clear in Fig 2, which ssuggest the CNN runs in parallel with Fast CNN. I got a different impression from the text.

2, These features may be used by:
- probabilistic rules obtained by what seems like a variation of a Priori.It woould be useful to have more  details on how  you compute the parameters. Also, you may want to look into softening to avoid 100% probabilities.

- User-written rules: not much detail is given, Table2 might help, f it was there ?

3. you use w2vec, bert and a singke-layer perceptron> Why these choices?

Results¨

- related systems: why these? How were they trained?

The english needs work, 3.4.2 is a particularly bad ex.

Finally, i was hoping, but I failed;
 to understand why your model works
 to see other applications

---

### Official Review · Reviewer_fQgQ · 2022-11-05

**Confidence:** 4
**Correctness:** 4
**Technical Novelty And Significance:** 1
**Empirical Novelty And Significance:** 1
**Recommendation:** 3

**Clarity, Quality, Novelty And Reproducibility:**

**Clarity**: The writing quality is ok but could be improved, but the paper is generally straightforward. Some sections of the paper could be reworked to become more focused. E.g. the 1-page approach overview in the modeling section, before diving into the architecture details, stands as some sort of a second introduction. It may be good to rework the presentation of this part together with the intro to reach the main ideas faster. The multiple diagrams and visualizations though are semantically clear and helpful to convey the ideas of the paper. Would be good to improve the resolution and some formatting mistakes in e.g. figure 3.

**Novelty**: The high-level idea of augmenting convolution with logical rules is a simple and nice idea in my opinion, that is novel as far as I’m aware (but not certain). From technical perspective, at the more concrete level the novelty of the paper is a bit limited.

**Reproducibility**: The paper provides hyperparameter selection and describes each component of the approach. The use of non-public data reduces the reproducibility though. I didn’t see a mention of releasing the constructed dataset either.

**Minor Comments**:
* **Spaces**: missing space on the second and fourth line of the introduction.
* **Citations**: I believe the wrong format is used for some of the citations (citet vs citep).
* **Subtitle**: on page 7 would be better to call the subsection title “Results” instead of “Result”.


**Strength And Weaknesses:**

**Strengths**:
* **Motivation and general idea**: The paper does well in motivating the potential benefits of combining learned classification with crispier logical rules and the introduction is completing and setting the right content and necessary background.
* **Results**: The proposed approach achieves improved results compared to the baseline.

**Weaknesses**:
* **Convoluted and heavy-engineered embedding approach**: The pipeline that combines object detection, BERT and word2vec makes the approach unnecessarily complicated and weakens the main idea of the paper, since it becomes unclear whether the empirical improvements in performance come from the approach itself or since we use multiple pretrained models that had access to further strong object-level and textual supervision. Would be good both to have more experiments to justify the contribution of each component, and look for ways to simplify the approach. The most obvious one is the use of both BERT and Word2Vec -- is that really necessary or maybe the reliance on two parallel approaches could be eliminated? Are there ways to maybe e.g. combine representations from different layers of BERT to avoid using also Word2Vec?
* **Usage of heuristics**: For instance, in the object-detection section, the paper uses a heuristic to avoid duplicated objects by counting them. But faster R-CNN gives many potentially overlapping predictions with different degrees of confidence. Extracting an object count using such heuristic may not work effectively. Likewise, the arbitrary probabilities given to objects, selected by a person, are unjustified, and the need to use rules made by hand might greatly limit the robustness and applicability of the approach to different domains and distributions
* **No use of standard datasets**: The paper uses a custom dataset rather than standard ones for a task where there could be plenty of public datasets that could be explored. This is a major weakness and this choice is unjustified in the paper either. The constructed dataset is also very small (733 images (440/146/146 for training/validation/testing)), weakening the statistical strength of the empirical results.


**Summary Of The Paper:**

A neuro-symbolic approach is proposed where a convolution neural network is extended with structured if-then symbolic rules based on word embeddings to improve image classification.

**Summary Of The Review:**

The paper explores a nice and simple idea, but the approach is potentially too complicated and heavy-engineered and the actual technical novelty is quite limited. This is more of an application paper than one that studies core new research. The experiments are also limited both in terms of dataset explored and baselines compared to. I therefore at this point unfortunately recommend rejection but encourage the authors to keep working on the paper to improve the discussed aspects.

---

### Decision · Program_Chairs · 2023-01-20

**Decision:**

Reject

**Justification For Why Not Higher Score:**

I am confident that this submission is not of ICLR quality.

**Justification For Why Not Lower Score:**

As described above.

**Metareview: Summary, Strengths And Weaknesses:**

This submission proposes a new approach to neural symbolic processing. Experiments are conducted on a non-standard dataset showing improvements over the baselines.

The four reviewers have a very consistent view of this submission.

Strength
* An important problem is studied.
* A new approach is proposed.

Weakness
* The proposed approach is too complicated and heavily engineered. There is no justification for each of the components. There are also heuristics used, which lack justification.
* The experimental results could be more satisfactory. The dataset was not standard. The baselines needed to be selected appropriately. There are stronger baselines.
* The significance of the work is limited. The authors need to convince the readers that the proposed approach works.
* Presentation can be further improved.